# Intra-Patient Evolution of HIV-2 Molecular Properties

**DOI:** 10.3390/v14112447

**Published:** 2022-11-04

**Authors:** Angelica A. Palm, Joakim Esbjörnsson, Anders Kvist, Fredrik Månsson, Antonio Biague, Hans Norrgren, Marianne Jansson, Patrik Medstrand

**Affiliations:** 1Department of Laboratory Medicine, Lund University, 22184 Lund, Sweden; 2Department of Translational Medicine, Lund University, 20502 Lund, Sweden; 3Nuffield Department of Medicine, University of Oxford, Oxford OX3 7BN, UK; 4Department of Clinical Sciences, Lund University, 22184 Lund, Sweden; 5National Public Health Laboratory, Bissau 1041, Guinea-Bissau

**Keywords:** HIV-1, HIV-2, evolution, disease progression, PNGS, coreceptor, molecular properties

## Abstract

Limited data are available on the pathogenesis of HIV-2, and the evolution of Env molecular properties during disease progression is not fully elucidated. We investigated the intra-patient evolution of molecular properties of HIV-2 Env regions (V1–C3) during the asymptomatic, treatment-naïve phase of the infection in 16 study participants, stratified into faster or slower progressors. Most notably, the rate of change in the number of potential N-linked glycosylation sites (PNGS) within the Env (V1–C3) regions differed between progressor groups. With declining CD4^+^ T-cell levels, slower progressors showed, on average, a decrease in the number of PNGSs, while faster progressors showed no significant change. Furthermore, diversity increased significantly with time in faster progressors, whereas no such change was observed in slower progressors. No differences were identified between the progressor groups in the evolution of length or charge of the analyzed Env regions. Predicted virus CXCR4 use was rare and did not emerge as a dominating viral population during the studied disease course (median 7.9 years, interquartile range [IQR]: 5.2–14.0) in either progressor groups. Further work building on our observations may explain molecular hallmarks of HIV-2 disease progression and differences in pathogenesis between HIV-1 and HIV-2.

## 1. Introduction

Human immunodeficiency virus type 1 (HIV-1) and 2 (HIV-2) are both causative agents of AIDS, although the progression to AIDS is slower in HIV-2 compared with HIV-1 infection [1,2]. The rate of CD4^+^ T-cell decline and disease progression is also slower in HIV-2 compared with HIV-1 infection, accompanied by a lower level of immune activation and more preserved and polyfunctional CD4^+^ and CD8^+^ T-cell responses [2,3]. Viral DNA load [4] and the plasma viral RNA load at comparable CD4^+^ T-cell counts are significantly lower in HIV-2 compared with HIV-1 infections [5,6,7]. This suggests that HIV-2 has a lower replication rate [8,9] or is more susceptible to immune control [3].

Both HIV-1 and HIV-2 diversity has been reported to increase over the course of infection [10,11,12], and the diversity rate to be positively correlated with the rates of CD4^+^ T-cell decline [11]. HIV-1 disease progression has also been shown to correlate with the evolution of coreceptor use and with glycosylation patterns, as well as the charge and length of Env regions [13,14,15,16,17,18,19]. Similar to HIV-1 infections, viruses with CXCR4 use have been isolated from HIV-2-infected individuals with low CD4^+^ T-cell counts or clinical symptoms of AIDS [20,21,22,23,24,25]. The V2 length and Env glycosylation density have been shown to increase during the asymptomatic phase of HIV-1 infection, likely as an attempt to escape the immune response [15,17,26,27,28]. Much less is known about the role of Env molecular properties in HIV-2 infection. However, since virus variants that have escaped neutralizing antibodies seem to be rare in HIV-2 infection [23,29,30], the evolution of Env may be less pronounced in HIV-2 compared with HIV-1 infection. Indeed, no clear trends for changes in potential N-linked glycosylation sites (PNGS) in Env over time have been reported for HIV-2 [11,23], but the V1–V2 regions have been suggested to be longer in patients with less advanced HIV-2 disease progression [23].

In a previous study, we stratified 16 HIV-2 infected individuals as faster or slower progressors based on longitudinal CD4^+^ T-cell dynamics and investigated the relationship between HIV-2 evolutionary rate in env and disease progression [31]. Faster disease progression, as determined by the level of CD4%, was found to be associated with an almost twice as high evolutionary rate compared with slower disease progression. In the present study, we investigated the intra-patient evolution of molecular properties during the asymptomatic, treatment-naïve phase of HIV-2 infection, using the same set of longitudinally obtained Env sequences as described above. We also investigated the evolution of molecular properties in relation to disease progression using the previous definition of HIV-2 progressor groups [31].

## 2. Materials and Methods

### 2.1. Study Population

The dataset in this study has been described in detail elsewhere [31] and was obtained from a large cohort of police officers in Guinea-Bissau, West Africa (described in detail in Appendix A, and [32,33]). The cohort was formed in 1990 and closed in 2011, but selected participants were asked to participate in a special sampling round, including a clinical examination and collection of a blood sample, in September 2013. A national antiretroviral therapy (ART) program was introduced to Guinea-Bissau in 2005, and the police cohort was included in early 2006. All participants had a clinical examination and the collection of a blood sample at each follow-up visit. Follow-up visits were scheduled every 12–18 months, but the actual average time between visits ranged between 15–32 months for the individuals included in this study (Appendix A).

In brief, 53 plasma samples from 16 longitudinally sampled HIV-2 infected individuals (two women and 14 men) were included and analyzed in this study. The inclusion criteria are presented in Appendix A. For the individuals included in this study, the median observation time from inclusion in the cohort until the last registered visit was 19.2 years (IQR: 15.0–20.8) and the median time from the first to the last amplified sample was 7.9 years (IQR: 5.2–14.0) [31]. The date of infection, defined as the midpoint between the last HIV-2 seronegative and the first seropositive sample, was known for seven individuals. The main objective of this study was to investigate the intra-patient evolution of HIV-2 during the asymptomatic, treatment-naïve phase of infection. However, seven samples from 5 individuals and 2 samples from 2 other individuals were collected after the individual had developed AIDS or initiated ART, respectively (Table 1). All AIDS cases were defined based on a CD4% < 14 or a CD4^+^ T-cell count < 200 cells/µL. No individual presented with clinical symptoms of AIDS.

The 16 HIV-2 infected individuals were previously stratified as faster or slower progressors using three different parameters [31] based on longitudinal CD4^+^ T-cell dynamics. CD4%, rather than CD4^+^ T-cell counts, was used in these studies as it was previously found to be more suitable in resource-limited settings due to a lower variability and sensitivity to specimen handling, age of individual and time of sampling [34,35,36]. Stratifications were based on: the CD4% decline rate, CD4% level and a combined coefficient (Table 1, Appendix A, and [31]). For each stratification, the mean of all HIV-2-infected individuals with two or more CD4% measurements in the cohort (n = 192) was determined, and faster and slower progression was determined based on a value above or below the mean [31]. The CD4% level and the combined coefficient stratifications resulted in identical groups (six slow progressors and 10 fast progressors, Table 1) but with different internal rankings. The CD4% decline rate stratification resulted in seven slow progressors and nine fast progressors. The median age at the first HIV-2 positive sample did not differ between groups (data not shown).

### 2.2. Amplification and Sequence Analysis

Details on amplification and sequence analysis are found in Appendix A, and [31]. In brief, plasma viral RNA was extracted, amplified, cloned, sequenced and manually edited. The analyzed regions were chosen based on their previous inclusion in studies on molecular properties in HIV-1 and HIV-2 infection [15,17,23,28,37]. Diversity analyses require aligned sequences, and regions that were difficult to align were, therefore, codon-stripped. The resulting fragment was 774 nucleotides long and spanned the last 30 nucleotides in the 3′ end of the C1 region, the entire V1–C3 regions and the first 6 nucleotides in the 5′ end of the V4 region. By contrast, analyses of PNGS, charge and length variation were performed using the full length of the translated amino acid sequences. Sub-regions of env were defined as follows (positions relative to HIV-2 reference sequence BEN, GenBank accession number M30502): V1–V2 = pos. 7040–7318; C2 = pos. 7319–7615; V3 = pos. 7616–7717; C3 = pos. 7718–7891. Analyses of PNGS, charge and length were performed using an in-house Perl script with PNGS defined as in N-glycosite [38]. The net charge of sequences was determined based on each lysine and arginine contributing +1 and each aspartic acid and glutamic acid contributing −1. Coreceptor tropism was predicted using the four major determinants method of dual/CXCR4 coreceptor use (L18Z, V19K/R, V3 net charge >+6, insertions at position 24), described previously [39,40]. CXCR4 use was considered when at least one of the criteria was fulfilled. Coreceptor tropism was also predicted using Geno2Pheno[coreceptor-hiv2] [41] with a false positive rate of 10%.

### 2.3. Phylogenetic Analysis

Nucleotide sequences were aligned with reference sequences of the major HIV-2 groups (downloaded from the Los Alamos Sequence Database [42]) in MEGA5 [43] using the Clustal algorithm, and a maximum-likelihood (ML) phylogenetic tree was reconstructed using Garli v2.0 [44] (using the inferred GTR+I+G substitution model). All sequences were of HIV-2 group A and displayed patient-specific clustering. The ML-based approximate likelihood ratio test (aLRT) Shimodaira-Hasegawa (SH)-like branch support was used to assess statistical support for internal branches, as implemented in PhyML 3.0 [45]. SH-values above 0.9 were considered statistically significant [46]. To investigate the temporal structure in HIV-2 evolution, ML phylogenetic trees were also reconstructed for each individual dataset separately and analyzed in Tempest (v 1.5.3, http://tree.bio.ed.ac.uk/software/pathogen, accessed on 22 June 2021 [47]) as described in [48].

### 2.4. Diversity Analysis

The diversity at each sample time-point was calculated by averaging the pairwise tree distance between all nucleotide sequences obtained from that time-point. We reconstructed 1000 bootstrap alignments and ML phylogenies in Garli v2.0 (GTR+I+G) [44], and the median diversity of the 1000 average pairwise diversity for each time-point was used in further analysis. Time points with less than three clones were excluded from the analysis (n = 2) since the uncertainty of diversity estimates is large on a small number of data points.

### 2.5. Statistical Analysis

To model the evolution of molecular properties (diversity, fragment length, fragment charge, and number of PNGS) over the course of infection and to test for differences between faster and slower progressors, we used linear mixed models. CD4% was used as a surrogate marker for disease progression because the time of infection could only be estimated for 7 of 12 patients. Each dependent variable (molecular property) was modeled as a function of CD4% as a continuous fixed effect and the progressor group (faster and slower progressors) as a categorical fixed effect. The interaction term between CD4% and the progressor group was included to allow for a different relationship between each dependent variable and CD4% in faster and slower progressors. Patient was included as a random effect, with both level (intercept) and effect of CD4% (slope) on the dependent variable allowed to vary between patients to model variance resulting from random differences in the evolution of molecular properties between virus population samples from different patients. If the model did not converge, a simpler random intercept model was used, excluding the patient random effect of CD4% (i.e., slope).

To investigate if there was a difference in the evolution of a molecular property (the mean slope or change in property) with a change in CD4% between progressor groups, we tested for the interaction of main effects (progressor group × CD4%). The main effect of the progressor group, i.e., the difference in mean intercept, is not biologically relevant in itself as the intercept represents the mean value of the molecular property at a CD4% of zero. Instead, we used the model to estimate and test for differences in mean values of each property at CD4% values corresponding to the time of infection (35%) and at AIDS onset (14%). The CD4% at the time of infection was conservatively estimated as the lower bound of the 95% confidence interval for the first available CD4% measurement for all HIV-negative individuals in the cohort (mean: 38.2%; confidence interval: 35.0–41.4%; n: 360; measurements from both negative controls and individuals subsequently infected; age at inclusions did not differ between controls and individuals include in this study). In addition to the main effects, we also tested if there was a significant effect of CD4% within each progressor group by testing if the model slope differed from 0. Due to a lack of variation between samples and individuals, mixed model analyses were not performed for C2 length, V3 length, V3 PNGS and C3 length.

We also modeled the evolution of the same molecular properties against the sampling date using mixed models. However, here we did not test for differences in the intercept or any other estimated means since the timing of sampling in relation to disease progression was not aligned between patients.

The distribution of PNGS over the different HIV-2 Env sequence regions was compared using Friedman’s test, followed by the Bonferroni corrected Wilcoxon signed rank test for pairwise comparisons of regions.

All statistical analyses were performed in IBM SPSS Statistics 21 (https://www.ibm.com/products/spss-statistics, accessed on 21 October 2021).

## 3. Results

### 3.1. Study Cohort and Stratification as Faster and Slower Disease Progressors

In this study, we included 409 HIV-2 V1–C3 Env sequences derived from 53 plasma samples of 16 longitudinally sampled HIV-2 infected individuals (two to five samples per individual), described in detail previously [31]. To study if the evolution of HIV-2 differed between disease progressor groups, individuals were stratified as faster or slower progressors according to the CD4% dynamics as previously described [31] (Table 1). Faster and slower progressors were defined based on three different stratifications; CD4% decline rate, CD4% level and a combined coefficient (Table 1, Appendix A, and [31]). Previously, we found a strong association between the evolutionary rate and the CD4% level and the combined coefficient stratifications but not with the CD4% decline rate [31]. Therefore, we present the results for the CD4% level and the combined coefficient (which resulted in identical groups) in the main manuscript. Results for the CD4% decline rate stratification are presented in Appendix A.

### 3.2. Phylogenetic Analyses and Genetic Diversity of the Env V1–C3 Regions

We investigated the temporal structure and sample-specific clustering of HIV-2 nucleotide sequences. A temporal structure was identified in all 16 individuals. In seven individuals, sequences from different time points formed segregated clusters (Appendix A). By contrast, in the remaining nine individuals, sequences from two or more time points were inter-mixed and did not form segregated clusters.

We found no difference in the rate of change in nucleotide sequence diversity with a change in CD4% between faster and slower progressors (*p* = 0.876, Table 2). Therefore, we simplified the model by assuming a common rate of change in both groups and excluded the interaction term. In the simplified model, we found no significant difference in the level of diversity between progressor groups (*p* = 0.822) and also no significant change in diversity with CD4% (*p* = 0.364).

Next, we analyzed the relationship between nucleotide sequence diversity and time. Faster and slower progressors differed significantly in the rate of change in diversity with time (*p* = 0.009, Table 3, Appendix A). For faster progressors, diversity increased significantly (*p* = 0.003), while for slower progressors, there was no significant change in diversity with time (*p* = 0.439).

### 3.3. Potential N-Linked Glycosylation Sites in the Env V1–C3 Regions

We investigated the PNGS pattern, i.e., the number and positions of aligned PNGS in the analyzed HIV-2 Env amino acid sequences. The V1–C3 regions had a median of 18 PNGS (IQR: 17–19), which were concentrated to the V1–V2 and C2 regions compared to V3 and C3 (*p* = 1.55 × 10^−9^, Friedmans test; pairwise comparisons between regions are presented in Appendix A). The median number of PNGS in the V1–V2 regions was seven (IQR: 6–8). Due to difficulties aligning the V1–V2 regions, the degree of conservation could not be reliably estimated. The C2 and V3 regions contained eight and one highly conserved PNGS, respectively (Table 1). The C3 region was the most variable region, both between and within individuals, with PNGS identified at 12 different sites, of which only one was conserved among sequences (Table 1). Although there were small variations in PNGS distribution within individuals, we did not, by visual inspection, observe any apparent patterns of evolution of PNGS distribution over time or with the development of AIDS or initiation of ART. Furthermore, faster and slower progressors could not be visually differentiated by the location of PNGS in the C2–C3 Env fragment.

Faster and slower progressors differed significantly in the rate of change in PNGS with a change in CD4% in the complete V1–C3 region (*p* = 0.033, Table 2, Appendix A). For slower progressors, the number of PNGS decreased significantly (*p* = 0.007), while for faster progressors, there was no significant change with decreasing CD4% (*p* = 0.822). Analyzing the component sub-regions separately, we found a significant difference in the rate of change in PNGS with CD4% in V1–V2 (*p* = 0.026) but not in C2 (*p* = 0.978) or C3 (*p* = 0.976). The V3 region was not analyzed due to the limited number and variability of PNGS between samples (Table 1). Within V1–V2, the pattern of change was the same as for the complete V1–C3 region; that is, for slower progressors, the number of PNGS decreased significantly (*p* = 0.005), while for faster progressors, there was no significant change with decreasing CD4% (*p* = 0.822, Figure 1 and Appendix A). For the C2 and C3 regions, there was no significant change in the number of PNGS with decreasing CD4% (*p* = 0.997 and *p* = 0.718, respectively) after removing the interaction term and analyzing the progressor groups jointly. Similar effect sizes were obtained when analyses for V1–C3 and V1–V2 were repeated, excluding samples collected after the onset of AIDS or initiation of ART, although statistical significance was not reached (Table 2).

We found no difference in the rate of change in PNGS with time (sampling date) between faster and slower progressors in the complete V1–C3 region (*p* = 0.273, Table 3). However, analyzing the component sub-regions separately, we did find a significant difference in the rate of change in PNGS with time for C3 (*p* = 0.003) but not for any of the other analyzed regions. For slower progressors, the number of PNGS in the C3 region increased significantly (*p* = 0.021), while for faster progressors, the number decreased significantly with time (*p* = 0.037).

### 3.4. Length and Charge Variation of the Env V1–C3 Regions

We also investigated the evolution of length and charge of the different sub-regions of Env V1–C3 amino acid sequences. As expected, V1–V2 demonstrated the largest variation in length both between and within individuals, and between time points, with a median of 94 amino acids, an IQR of 91–97 and a range of 74–120. Minor variations in length were also observed in C3, but only between individuals (median = 58 amino acids, IQR: 58–58, range: 56–61). The length of C2 and V3 was conserved at 99 and 34 amino acids, respectively, in all 409 sequences.

Faster and slower progressors did not differ significantly in the rate of change in length with change in CD4% for the complete V1–C3 region (*p* = 0.112, Table 2) or for the V1–V2 sub-region (*p* = 0.119). In a simplified model, excluding the interaction term, there was no significant change in length with a change in CD4% and no significant difference in mean length between progressor groups for V1–C3 or for V1–V2 (Table 2). The C2, V3 and C3 sub-regions were not analyzed because of limited variation (see above). Similarly, we found no significant difference in the rate of change of length with time between the progressor groups or, after excluding the interaction term, any significant change in fragment length with time for the complete V1–C3 region or for any of the analyzed sub-regions (Table 3).

Next, we investigated the estimated charge of the fragments. We found no difference between progressor groups in the rate of change in charge with CD4% for the complete V1–C3 region or for any of the sub-regions (Table 2). In the simplified model with the interaction term removed, the charge of the complete V1–C3 region did not change significantly with CD4% (*p* = 0.178). However, the V1–V2 sub-region did show a significant increase in charge with decreasing CD4% (*p* = 0.017, Table 2). For the other regions, the charge did not change significantly with CD4%, and there was no significant difference in the mean charge between faster and slower progressors for any analyzed region (Table 2). Finally, we found no significant difference in the rate of change of charge with time between faster and slower progressors, or, when the interaction term was removed, any significant change in charge with time for the complete V1–C3 region or for any of the analyzed sub-regions (Table 3).

### 3.5. Predicted Coreceptor Use

We predicted coreceptor use based on the viral sequences included in this study using two different methods: the four major determinants method of dual/CXCR4 use (L18Z, V19K/R, V3 net charge >+6, insertions at position 24) [39,40]; and Geno2Pheno[coreceptor-hiv2] [41]. Both methods predicted that the vast majority of the V3 sequences derived from viruses using CCR5, and not CXCR4, and only one and eight sequences were predicted to derive from viruses using CXCR4 by the Geno2Pheno and the four major determinants method, respectively (Table 4). The seven sequences where viruses were predicted to use CXCR4 by the four major determinants method and not by Geno2Pheno, were still reported to possibly use CXCR4 by Geno2Pheno due to the net charge of +7, although this criterion itself is not sufficient for CXCR4 prediction by Geno2Pheno. The eight sequences from viruses predicted to use CXCR4 came from four different samples, each from a different individual (Table 4). All four samples also included sequences where viruses were predicted to use CCR5, and for three of the samples, a minority of sequences were predicted to use CXCR4 (Table 4). The only sample from which the majority of sequences were from viruses predicted to use CXCR4 (four of five sequences) was from an individual that developed AIDS (CD4^+^ T-cell count = 138 cells/µL, CD4% = 17) at the time-point when this sample was taken. However, CXCR4 use was not identified from later time points in any of the four individuals, including the individual who had developed AIDS and later presented with very low T-cell levels (CD4^+^ T-cell count = 60 cells/µL, CD4% = 6). The individuals with viruses predicted to use CXCR4 were from both progressor groups.

## 4. Discussion

In a previous study, we stratified 16 treatment-naïve individuals into faster and slower progressors based on the level of CD4% and found a strong association between the evolutionary rate and the CD4% level [31]. In the present study, we used the same stratification to investigate the evolution of molecular properties of HIV-2 Env regions (V1–C3) over the asymptomatic phase of infection.

Faster and slower progressors differed significantly in the rate of change in the number of PNGS in V1–C3 with decreasing CD4%. The difference was significant for V1–V2 but not for other sub-regions, indicating that PNGS in V1–V2 may be driving the divergent patterns. On average, the number of PNGS decreased in slower progressors, whereas faster progressors maintained a near-constant level with decreasing CD4%. This finding is in contrast to what has been observed during HIV-1 infection, where Env has been shown to evolve towards a denser glycan shield during the asymptomatic phase of the infection [17,26,27,28], only to lose glycans in end-stage disease [13]. For HIV-1, loss of PNGS has been associated with increased virus infectivity but also with increased neutralization sensitivity [13,27,49,50]. Thus, accumulation of PNGS may represent a way for the virus to evade the immune system at the cost of virus infectivity. The observed difference in PNGS evolution between HIV-2 faster and slower progressors could possibly relate to differences in the level of antigen stimulation, i.e., virus replication, which in turn influences the magnitude of the virus-specific immune response. HIV-1 elite controllers, a group that in large resembles the majority of HIV-2 infected individuals in terms of plasma viral load and level of immune activation, have been reported to have fewer glycans in Env compared with chronic HIV-1 progressors or ART-treated individuals [51]. Together, these findings may suggest that slower progressor groups (e.g., HIV-1 elite controllers and HIV-2 slower progressors) do not exert an immune pressure sufficient to drive the selection of viral variants with increased levels of PNGS.

We also performed analyses of HIV-2 nucleotide sequence diversity and segregation. Similar to previous reports of HIV-2 [10,11], we found that both segregated clusters and the intermixing of sequences from earlier and later time points were common. HIV-2 diversity has been suggested to increase from earlier to later time points [10,11,52,53]. Indeed, we found that diversity increased over time in faster progressors. However, in slower progressors, diversity did not change significantly with time, and the estimated mean even showed a slight decrease, possibly indicating a lower viral turnover in slower compared to faster progressors. Investigations on the association between diversity and viral load would be interesting. However, viral load measurements have not been included as a standard procedure in the study cohort in Guinea-Bissau. The generally low viral load in HIV-2-infected individuals presents a large technical challenge, especially in the analysis of sequences obtained from plasma viral RNA. The low viral load also poses a potential risk of generating few viral clones and possibly undersampling of quasispecies. In addition, while we address differences between relatively faster and slower HIV-2 progressors, aviremic individuals could not be included in this study based on plasma viral RNA. Still, the low detection limit of our assay (12 RNA copies/mL plasma [31]) allowed for the inclusion of individuals with very low RNA levels. Furthermore, the analysis of viral RNA from plasma allowed us to study the actively replicating virus populations, which may not be detected in analysis based on proviral DNA, or RNA propagated in the virus.

In this study, we used CD4% as a surrogate marker of disease progression since the estimated date of infection was only known for seven of the individuals. However, we also investigated the evolution of HIV-2 over time during the asymptomatic phase of infection. Compared to HIV-1 infection, the CD4% has been shown to be higher in early HIV-2 infection with a slower subsequent decline in CD4 [2]. In addition, HIV-2-infected individuals have been shown to develop clinical AIDS at a higher CD4% compared with HIV-1-infected individuals [2]. We found that some of the evolutionary signatures differed when we analyzed the HIV-2 molecular properties in relation to CD4% decline (i.e., disease progression) and in relation to time (sampling date). Notably, the number of PNGS in the V1–V2 and V1–C3 regions and the charge of the V1–V2 regions were associated with disease progression (i.e., CD4%), while the number of PNGS in the C3 region and diversity showed associations with time. We have previously shown that HIV-2 env is under overall negative selection [31], which in part may explain the increased diversity over time as neutral variation is allowed to accumulate. In contrast, the evolution of PNGS in the V1–V2 regions and the entire V1–C3 fragment may represent adaptations associated with CD4% decline and disease progression. As the immune pressure differs at different stages of disease progression, it is possible that the PNGS will adjust accordingly to maintain the balance between preventing immune system recognition while still enabling efficient binding to and infection of target cells [27,50]. Thus, we speculate that certain traits, such as the evolution of PNGS, may be adaptively associated with disease progression, whereas other traits, such as diversity, largely represents accrued neutral variation in a region under negative selection.

The associations identified in this study, with time as well as with CD4%, were found to be located in the V1–V2 and C3 regions. Furthermore, several non-conserved PNGS were identified in the C3 region. Our results support observations from previous studies, suggesting that the majority of genotypic and phenotypic evolution of the V1–C3 regions are accounted for by the V1–V2 and C3 regions [10,31,37]. Both the C2 and C3 regions have previously been found to be well exposed, while the V3 region is concealed within the envelope complex [37]. We confirmed the location of six conserved PNGS in HIV-2 Env (four in C2, one in V3 and one in C3 [10,37]), and due to the inclusion of the complete C2 region in this study, we identified four additional conserved PNGS in the N-terminal part of the C2 region. A deeper understanding of the evolutionary constraints of the different regions could provide important clues that may guide future vaccine development. In this regard, it would be valuable to expand the coverage of the analyzed regions. We investigated the C1–V3 region but important genetic and phenotypic traits linked to evolution and disease progression may also localize to other parts of the genome and thus be missed in this study.

Although our results are in line with previous studies reporting a relatively low evolution of HIV-2 molecular properties [10,11,23], with few differences identified between faster and slower progressors, the lack of statistical power poses an important limitation of this study. The lack of identified associations and differences between progressor groups may, in part, be a reflection of the limited size of the dataset. Still, with the longitudinal sampling of 16 HIV-2 infected individuals over a median of 7.9 years, this is, to the best of our knowledge, the most comprehensive study on the evolution of molecular properties in HIV-2 infected treatment-naïve individuals. To draw definitive conclusions regarding the evolution of HIV-2 molecular properties in general and between specific progressor groups, a larger study population would likely be required. In this study, we included a few samples collected after the individual had developed AIDS or initiated ART. While we verify that the effect differences we find between progressor groups remain when these samples are excluded, the statistical significance is lost. This illustrates a limitation of our relatively small dataset, and the validation of our data should focus on samples collected during the asymptomatic stage of infection. Future studies should preferably also cover a larger fragment of the genome, as regions outside of the V1–C3 region of env included here may provide additional information about genotypic and phenotypic evolution in relation to disease progression. Still, our data point towards a relatively slow evolution of diversity, fragment length, fragment charge and PNGS over time and with declining levels of CD4% in the V1–C3 regions of HIV-2. Despite the limitations, we identify differences between faster and slower progressors in terms of diversity and the number of PNGS. The stratification of HIV-2-infected individuals into different progressor groups in the study of the evolution of molecular properties has not been previously reported. As such, our findings bring new insights to the understanding of the attenuated pathogenesis of HIV-2.

We predicted the coreceptor tropism of the viruses, from which sequences were derived in this study, using two different prediction methods [39,40,41]. Only eight virus sequences were predicted to use CXCR4, with no differences between progressor groups. The low frequency of CXCR4 use was not surprising as virus CXCR4 use has previously only been reported from HIV-2 infected individuals with clinical symptoms of AIDS or low CD4^+^ T-cell counts [20,21,22,23,24,25]. However, our results present the first observation, based on longitudinal plasma viral RNA samples, that sequences predicting virus CXCR4 use are rare during the asymptomatic phase of HIV-2 infection. Notably, the few sequences that predicted viruses with CXCR4 use appeared merely as transient viruses that did not subsequently emerge as a dominating viral population. In this study, we did not validate our predictions using functional assays, which would be valuable in future studies. It would also be interesting to follow a larger number of HIV-2-infected individuals throughout the entire disease course to see if and when a progressive switch in coreceptor use can be identified in the circulating virus population.

## 5. Conclusions

We investigated HIV-2 intra-patient evolution of Env molecular properties using sequences obtained from plasma viral RNA of 16 study participants with long follow-up over the asymptomatic, treatment-naïve phase of the infection. Our results suggest a generally limited evolution of molecular properties over time and with declining CD4%, with a few differences identified between faster and slower disease progressors. Diversity increased with time in faster progressors, while no such change was observed in slower progressors, and the number of PNGS in V1–V2 decreased with declining CD4% in slower progressors, whereas faster progressors showed no such change. Our results support previous work suggesting that the HIV-2 Env V1–V2 and C3 regions are more immunologically exposed than other regions [10,31,54]. Furthermore, the emergence of viruses using CXCR4 was rare. Studies of larger sample sizes are warranted to confirm our results and elucidate the role of molecular properties in HIV-2 disease progression. Our findings provide new insights into HIV-2 evolution, and studies building on these data could reveal important predictors of HIV-2 disease progression and enhance our understanding of HIV-2 pathogenesis.

## Figures and Tables

**Figure 1 viruses-14-02447-f001:**
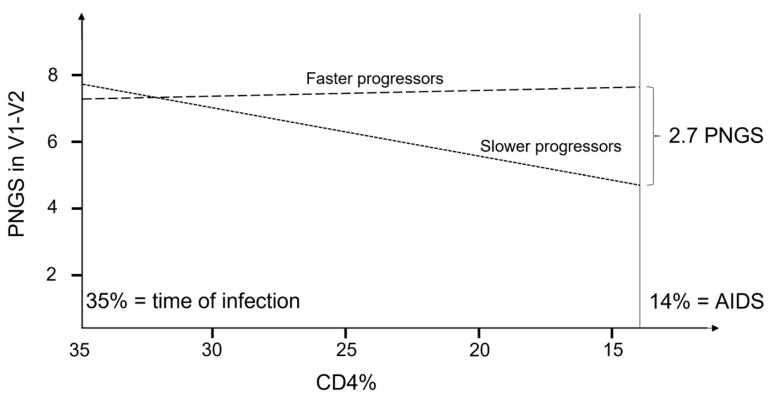
Schematic illustration of the evolution of PNGS as a function of CD4% for faster and slower progressors in the V1–V2 region. *p*-values for differences in slope between progressor groups was 0.014, and the *p*-values to test if the slope was different from zero were 0.007 and 0.822 for slower and faster progressors, respectively.

**Table 1 viruses-14-02447-t001:** Study population characteristics and distribution of potential N-linked glycosylation sites.

Individual	Sample	CD4%	CD4abs	AIDS ^a^	ART ^a^	Clones ^b^	Stratification ^c^		C2 ^d^									V3 ^d^	C3 ^d^												Total nr PNGS ^e^	
							1	2 & 3	206	238	241	248	272	278	289	296	300	310	343	357	365	366	367	368	369	371	371.25	371.75	375	389	C2–C3	V1–C3
DL3405	2001-10-20	37	798	N	N	11	Slower	Slower												R											11	20
	2004-06-26	25	198	N	N	8														R											11	20.5
	2008-10-07	46	782	N	N	9														R											11	21
DL3542	1994-11-07	35	235	N	N	9	Faster	Slower												K											10	17
	2013-09-25	36	n/a	Y	N	8														E											11	20
DL2051	1996-02-12	33	253	N	N	7	Faster	Slower																							11	16
	1997-09-26	38	613	N	N	5																									10	20
	2002-07-31	20	179	N	N	11																									10	13
	2006-11-28	33	369	N	N	5																									11	16
	2009-10-07	22	386	N	N	2																									11	16
DL2876	1995-12-28	35	559	N	N	6	Faster	Slower																							10.5	17
	1996-12-19	30	594	N	N	6																									10	17
	2000-08-23	23	282	N	N	9																									11	18
DL3654	1998-02-04	25	592	N	N	7	Slower	Slower																							11	17
	2013-09-27	29	n/a	N	N	7																									11	19
DL2533	1991-04-19	n/a	n/a	N	N	10	Slower	Slower																							11	20
	2001-12-04	29	515	N	N	7																									11	18
	2002-07-24	25	300	N	N	4																									10.5	16
	2006-11-14	32	611	N	N	12																									12	18
DL2316	2001-10-21	23	305	N	N	10	Faster	Faster																							10	18.5
	2002-07-22	26	161	Y	N	7																									10	19
DL2794	1993-11-10	11	116	N	N	10	Slower	Faster																							10	18
	1996-04-08	25	330	N	N	7																									11	20
	2004-06-26	13	91	Y	N	5																									10	18
DL3941	2004-12-08	n/a	n/a	N	N	12	Faster	Faster																							11	19
	2008-09-18	23	386	N	N	5																									10	16
	2010-03-10	18	267	N	N	8																									10.5	15
	2013-09-23	13	n/a	Y	N	6																									10	14
DL2381	2003-03-13	32	1156	N	N	7	Faster	Faster																							11	18
	2004-06-10	23	n/a	N	N	11																									11	18
	2009-09-28	15	253	N	N	12																									11	19
DL2335	2002-07-16	21	192	N	N	9	Faster	Faster																							10	17
	2003-11-14	20	240	N	N	12																									10	16
	2004-11-05	21	295	N	N	12																									8	14
	2007-10-11	n/a	n/a	N	N	7																									9	15
	2009-09-29	25	196	Y	N	2																									10	17.5
DL3647	2001-06-25	25	318	N	N	9	Faster	Faster																							9	17
	2006-11-11	17	138	Y	N	5																									8	20
	2010-03-19	6	60	Y	N	7																									10	20
DL3646	1996-04-08	21	670	N	N	9	Slower	Faster												K											11	19
	2006-11-10	17	373	N	N	6														K											10	18
	2008-10-04	14	326	N	N	9														K											10	18
	2009-10-19	n/a	n/a	N	N	10														K											10	18
	2010-05-11	20	231	N	Y	6														K											10	18
DL3222	1997-02-06	17	748	N	N	9	Slower	Faster												K											11	18
	2001-07-26	25	702	N	N	11														K											10	16
DL3740	2007-10-24	16	352	N	N	7	Slower	Faster																							10	16
	2009-03-30	17	318	N	N	8																									11	19
	2009-10-23	n/a	n/a	N	N	4																									11	18.5
	2013-09-16	16	n/a	N	Y	9																									10	17
DL2386	2008-10-16	n/a	n/a	N	N	5	Faster	Faster																							10	17
	2010-02-19	16	280	N	N	6																									11	18
	2013-09-26	15	n/a	N	N	4																									11	18
					Percentage of clones with PNGS				98.5	99.3	88.5	97.1	99.3	99.0	95.8	7.6	89.0	96.1	5.6	1.0	88.3	5.6	0.5	2.4	25.2	28.6	4.6	0.2	0.2	0.2		

PNGS present in black boxes > 80% of the sequences; dark grey boxes > 30% ≤ 80% of the sequences; light grey boxes ≤ 30% of the sequences. ^a^ Samples collected after the study participant developed AIDS or initiated ART. N = no, Y = yes. ^b^ The number of sequences included in the study from each sample. ^c^ Stratification of faster and slower progressors based on: 1. CD4% decline rate; 2. CD4% level; 3. Combined coefficient. ^d^ Position of PNGS in the C2, V3 and C3 regions, numbered according to position in Env of reference strain BEN (Accession number = M30502). Positions 371.25 and 371.75 represents amino acid insertions not present in BEN. ^e^ The median number of PNGS in the C2–C3 region and the V1–C3 region of each sequence. n/a—not available.

**Table 2 viruses-14-02447-t002:** Mixed model estimates of mean diversity, PNGS, fragment length and fragment charge at a CD4% level corresponding to 35% and 14% and mean rate of change of these properties with change in CD4% (slope) in faster and slower progressors.

	Slope ^a^			Slope ^b^		CD4% = 35 ^c^			CD4% = 14 ^c^		
Property	Slower	Faster	*p*	All Individuals	*p*	Slower	Faster	*p*	Slower	Faster	*p*
Diversity	−2.94 × 10^−4^	−4.18 × 10^−4^	0.876	−3.59 × 10^−4^	0.364	0.012	0.013	0.822	0.019	0.021	0.822
V1-C3 PNGS	0.149	−0.011	0.033 *	N/A	N/A	18.528	17.440	0.316	15.389	17.682	0.049 *
V1-C3 PNGS ^d^	0.143	−0.005	0.089	N/A	N/A	18.355	17.381	0.432	15.355	17.495	0.073
V1-V2 PNGS	0.141	−0.010	0.026 *	N/A	N/A	7.718	7.270	0.662	4.747	7.489	0.014 *
V1-V2 PNGS ^d^	0.145	−0.002	0.055	N/A	N/A	7.604	7.145	0.666	4.565	7.179	0.013 *
C2 PNGS	−2.28 × 10^−4^	4.28 × 10^−4^	0.978	4.60 × 10^−5^	0.997	7.886	7.729	0.514	7.885	7.728	0.514
V3 PNGS ^e^	-	-	-	-	-	-	-	-	-	-	-
C3 PNGS	0.004	0.003	0.976	0.004	0.718	1.921	1.533	0.121	1.843	1.455	0.121
V1-C3 length	0.162	−0.206	0.112	0.007	0.957	284.916	284.792	0.965	284.764	284.640	0.965
V1-V2 length	0.152	−0.202	0.119	0.003	0.978	93.995	93.674	0.897	93.922	93.602	0.897
C2 length ^e^	-	-	-	-	-	-	-	-	-	-	-
V3 length ^e^	-	-	-	-	-	-	-	-	-	-	-
C3 length ^e^	-	-	-	-	-	-	-	-	-	-	-
V1-C3 charge	−0.056	−0.062	0.952	−0.059	0.178	7.319	6.695	0.519	8.567	7.943	0.519
V1-V2 charge	−0.087	−0.050	0.511	−0.068	0.017 *	−3.154	−4.417	0.093	−1.719	−2.982	0.093
C2 charge	0.027	0.037	0.805	0.032	0.109	3.596	3.842	0.623	2.929	3.174	0.623
V3 charge	−0.013	0.011	0.272	−0.001	0.931	5.064	5.612	0.167	5.084	5.632	0.167
C3 charge	0.013	−0.099	0.131	−0.053	0.166	1.386	1.192	0.782	2.493	2.299	0.782

^a^ Results of mixed model analysis including the interaction between CD4% and progressor group, i.e., the model includes a separate slope for the relationship between the property and increasing CD4% for each group. The *p*-value refers to the test of the interaction, i.e., if the relationship between the property and CD4% differs between groups. ^b^ Results of mixed model analysis excluding the interaction between CD4% and progressor group, i.e., the model estimates a common slope for the linear relationship between property and increasing CD4% for all individuals of both groups. This analysis was only performed if the interaction was not significant. The *p*-value refers to the test of the common slope versus zero, i.e., if the property changes significantly with decreasing CD4%. ^c^ Mixed model estimated means for the property at CD4% = 35 and CD4% = 14 for each progressor group. The *p*-value refers to the test of differences in the mean between groups. If the interaction term for the complete model was significant, reported values are from the complete model. If the interaction was not significant, the reported values are from the simplified model without the interaction. In the latter case, the net difference at both CD4% levels, and the *p*-values for the tests, will be identical. ^d^ Analysis performed with the exclusion of samples collected after AIDS onset or initiation of ART. ^e^ Analysis not performed due to limited variation of that property between samples. * Denotes significant *p*-value. N/A—not applicable.

**Table 3 viruses-14-02447-t003:** Mixed model estimates of the mean rate of change per year of diversity, PNGS, fragment length and fragment charge with sampling date (slope) in faster and slower progressors.

	Slope ^a^			Slope ^b^	
Property	Slower	Faster	*p*	All individuals	*p*
Diversity	−4.15 × 10^−4^	1.54 × 10^−3^	0.009 *	N/A	N/A
V1-C3 PNGS	3.24 × 10^−2^	−6.09 × 10^−2^	0.273	−1.17 × 10^−2^	0.782
V1-V2 PNGS	−6.86 × 10^−3^	−3.58 × 10^−2^	0.709	−2.00 × 10^−2^	0.602
C2 PNGS	5.97 × 10^−3^	7.54 × 10^−5^	0.801	3.20 × 10^−3^	0.781
V3 PNGS ^c^	-	-	-	-	-
C3 PNGS	3.15 × 10^−2^	−3.23 × 10^−2^	0.003 *	N/A	N/A
V1-C3 length	1.42 × 10^−1^	1.25 × 10^−1^	0.945	1.36 × 10^−1^	0.257
V1-V2 length	1.44 × 10^−1^	9.17 × 10^−2^	0.827	1.23 × 10^−1^	0.295
C2 length ^c^	-	-	-	-	-
V3 length ^c^	-	-	-	-	-
C3 length ^c^	-	-	-	-	-
V1-C3 charge	−2.49 × 10^−2^	3.39 × 10^−2^	0.551	−1.48 × 10^−3^	0.976
V1-V2 charge	1.51 × 10^−2^	1.13 × 10^−1^	0.137	5.39 × 10^−2^	0.109
C2 charge	5.96 × 10^−4^	−1.96 × 10^−2^	0.613	−7.92 × 10^−3^	0.685
V3 charge	−7.93 × 10^−3^	−3.25 × 10^−2^	0.347	−1.83 × 10^−2^	0.158
C3 charge	−8.00 × 10^−4^	3.42 × 10^−2^	0.574	1.51 × 10^−2^	0.624

^a^ Results of mixed model analysis including the interaction between time and progressor group, i.e., the model includes a separate slope for the relationship between the property and time for each group. The *p*-value refers to the test of the interaction, i.e., if the relationship between the property and time differs between groups. ^b^ Results of mixed model analysis excluding the interaction between time and progressor group, i.e., the model estimates a common slope for the linear relationship between property and time for all individuals of both groups. This analysis was only performed if the interaction was not significant. The *p*-value refers to the test of the common slope versus zero, i.e., if the property changes significantly with time. ^c^ Analysis not performed due to limited variation of that property between samples. * Denotes significant *p*-values. N/A—not applicable.

**Table 4 viruses-14-02447-t004:** Virus sequences predicted to use CXCR4.

				Prediction by Major Determinants ^a^		Prediction by Geno2Pheno ^b^	
Individual	Sample	CD4 Count	CD4%	Fulfilled Criteria for CXCR4 Use	Fraction of CXCR4 Using Sequences	X4-Capable	Fraction of CXCR4 Using Sequences
DL2533	2001-12-04	515	29	V3 charge +7	1/6	No ^c^	0/6
DL2876	1996-12-19	594	30	L18S	1/7	Yes	1/7
DL3647	2006-12-11	138	17	V3 charge +7	4/5	No ^c^	0/5
DL3740	2007-10-24	352	16	V3 charge +7	2/7	No ^c^	0/7

^a^ Dual/CXCR4 coreceptor use predicted when at least one of the following criteria was fulfilled: L18Z, V19K/R, V3 net charge >+6, insertions at position 24. ^b^ Geno2Pheno analysis were performed with a false positive rate of 10%. ^c^ Geno2Pheno does not classify viral sequences as X4-capable based on net charge alone. But a net charge exceeding +6 indicates that the virus might use the CXCR4 coreceptor, as reported in the Geno2Pheno output files.

## Data Availability

Sequences included in this study have been deposited in GenBank and assigned the following accession numbers: KM390990-KM391398.

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
