# Peer review of "Intra-Patient Evolution of HIV-2 Molecular Properties"

_viruses, 2022, doi:10.3390/v14112447_

Round 1
Reviewer 1 Report
The aim of the study was to evaluate how the molecular properties of HIV-2 Env V1-C3 regions evolve during disease progression. The molecular properties include genetic diversity, potential N-linked glycosylation sites (PNGS), length and charge variation and predicted coreceptor use. A total of 16 study participants at treatment-naïve phase were studied, who were then stratified into faster or slower progressors based on the CD4% measurement. The results suggested several differences between the faster and slower progressors during the evolution of HIV-2 Env V1-C3 regions: 1) Genetic diversity increased with time in faster progressors, but not in slower progressors; 2) PNGS in V1-V2 sub-regions decreased in slower progressors with reducing CD4%, but this was not observed in faster progressors; 3) The length and charge of Env regions showed no significant differences; 4) The predicted coreceptor use of CXCR4 was not a dominating viral population. The sample sizes are relatively small for the considerable variation in the measure of interest (i.e. differences in HIV-2 molecular properties) and likely variability in immune pressures on HIV-2 evolution between individuals. Nevertheless, the results and conclusions appear reasonable, and the topic of the manuscript is of interest to HIV researchers in general.
Major concern:
1. The statistics and model results are presented in tables. For a proper evaluation it is essential that the data (individual data points) and models are presented in plots for visualization.
Minor concerns :
1. HIV-2 Env contains several subunits. In this paper, only V1-C3 sequences were included in the analysis for evolution, while the other subunits were not mentioned. The authors should include a statement in the methods indicating the reason for the limited sequence coverage and include a discussion of what could be missed in the absence of better coverage.
2. Since the sample size in each category was very small (six slow progressors and 10 fast processors), the selection pressure could be different in the individuals. Hence it is suggested that the authors address the possibility of confounding effects from individual variations in selection pressure.
3. In line # 120, it was indicated that diversity analysis required perfectly aligned sequences. It is not well defined what ‘perfectly aligned’ means. It is suggested that the authors add in an explanation for how they define ‘ perfectly aligned ’ and explain why this is essential.
Reviewer 2 Report
The authors present an analysis of different Env regions from HIV-2 infected individuals. The study participants are classified as fast or slow progressors. The sampling of plasma viral RNA is over a long period of time (~20 years). This is an adequate time frame to study virus evolution. Unfortunately viral load readings were not available, however CD4 counts could be used to calculate CD4% as a marker of disease status. The authors were able to determine differences in Env diversity and potential N-linked glycosylation patterns in specific regions of Env between fast and slow progressors. The data generated is additive to the relatively understudied pathogenesis of HIV-2 disease and is relevant to the topic of viral evolution in the face of immune pressures. The manuscript, although possibly a bit long, is well written in the English language. The statistical analyses are extensive and although I’m not well-versed in the methods used I trust that the analyses were properly conducted. I recommend publication of the manuscript with a few suggestions that the authors may wish to address in the current version. All of the best with future research in the HIV-2 field.
Thank you.
2.1 study population
Line 70: collection of a blood sample (remember the plasma fraction has to be processed after collecting a blood sample from the donor)
Line 73: collection of a blood sample (delete “plasma”)
Same error in supplementary file Line 8 and 13 collection of a blood plasma sample (delete “plasma”) Line 18 can be left unchanged or “blood” can be deleted if assuming that the plasma has been processed and stored for retrospective analysis.
Table 1 DL3405 “Slower” under stratification 1 is not aligned in the centre of the box
Table 1 is already very busy but it might be useful to also have a “CD4+ T-cell count” column next to the CD4% column. If viral loads are available this would also be useful to include in the table – could retrospective viral load testing be applied to stored plasma samples?
There are feint grey lines appearing in the black boxed areas that need to be made black – probably caused by grey lines of the borders of the boxes.
Line 86: However, seven samples from five individuals and two samples from two other individuals were collected after the individual had developed AIDS or initiated ART, respectively (Table 1). Do you need to justify why viral sequences that were obtained after AIDS onset and after commencing ART were included in the analysis? Viral evolution is expected to decline when immunity wanes during AIDS development and similarly after ART initiation viral replication is reduced and viral evolution is also expected to decline.
Line 96 and line 34 in supplementary: For each stratification, the mean of all HIV-2-infected individuals with two or more CD4% measurements in the cohort (n=192) was determined and faster and slower progression was determined based on a value above or below the mean [31].
Can you check whether it was the median value for the cohort that was used as the cutoff for fast or slow progression not the mean?
2.2 Amplification and sequence analysis
In this section it is not always clear when nucleotide sequences or translated amino-acid sequences are being used for the analysis. This section could be made easier to follow if it is stated what type of “sequence” was used. This is also a problem in the results section eg genetic diversity versus protein diversity. It would be good to specify whether genetic or protein sequences are under discussion. Also check for consistent use of “env” for nucleotide sequence and “Env” for protein sequences referrals. Some of the headings are in italic font so this might be confusing.
Table 2 and 3: Could the p-values that were significant be marked in some way? Bold or asterisk etc.
Discussion:
Line 386: do not exert an immune pressure (exert is probably a better word to use than “harbor”)
Line 388: “diversity” Make it more specific that this is a discussion about genetic diversity to avoid confusion with protein amino-acid variability.
It might be good to include more points of limitations in the study. Small sample sizes. Low viral loads meant difficulty in amplification of viral RNA hence fewer clones were generated which means undersampling may have missed additional quasispecies. Predominantly male participants (14 vs 2) so sex differences cannot be addressed. Co-receptor use prediction could not be verified with functional assays. Other HIV-2 proteins were not analysed eg. Gag. HLA driven adaptations were not studied for Env.
